# An Eco-Friendly Method to Get a Bio-Based Dicarboxylic Acid Monomer 2,5-Furandicarboxylic Acid and Its Application in the Synthesis of Poly(hexylene 2,5-furandicarboxylate) (PHF)

**DOI:** 10.3390/polym11020197

**Published:** 2019-01-23

**Authors:** Junhua Zhang, Qidi Liang, Wenxing Xie, Lincai Peng, Liang He, Zhibin He, Susmita Paul Chowdhury, Ryan Christensen, Yonghao Ni

**Affiliations:** 1BiomassChem Group, Faculty of Chemical Engineering, Kunming University of Science and Technology, Kunming 650500, China; labaka1993@163.com (Q.L.); penglincai8@163.com (L.P.); heliangkmust@163.com (L.H.); 2Department of Chemical Engineering, University of New Brunswick, Fredericton, NB E3B 5A3, Canada; zhe@unb.ca (Z.H.); Susmita.paul@unb.ca (S.P.C.); Ryan.christensen@unb.ca (R.C.); 3Key Laboratory of Advanced Textile Materials and Manufacturing Technology of Ministry of Education, Zhejiang Sci-Tech University, Hangzhou 310018, China; 18720974664@163.com

**Keywords:** biomass-based polyester, 5-hydroxymethylfurfural, 2,5-furandicarboxylic acid, copolyester, poly(hexylene 2,5-furandicarboxylate)

## Abstract

Recently, we have developed an eco-friendly method for the preparation of a renewable dicarboxylic acid 2,5-furandicarboxylic acid (FDCA) from biomass-based 5-hydroxymethylfrufural (HMF). In the present work, we optimized our reported method, which used phosphate buffer and Fe(OH)_3_ as the stabilizer to improve the stability of potassium ferrate, then got a purified FDCA (up to 99%) in high yield (91.7 wt %) under mild conditions (25 °C, 15 min, air atmosphere). Subsequently, the obtained FDCA, along with 1,6-hexanediol (HDO), which was also made from HMF, were used as monomers for the synthesis of poly(hexylene 2,5-furandicarboxylate) (PHF) via direct esterification, and triphenyl phosphite was used as the antioxidant to alleviate the discoloration problem during the esterification. The intrinsic viscosity, mechanical properties, molecular structure, thermal properties, and degradability of the PHFs were measured or characterized by Koehler viscometer, universal tensile tester, Nuclear Magnetic Resonance (NMR), Fourier-transform Infrared (FTIR), X-ray diffraction (XRD), Differential Scanning Calorimeter (DSC), Derivative Thermogravimetry (DTG), Scanning Electron Microscope (SEM), and weight loss method. The experimental evidence clearly showed that the furan-aromatic polyesters prepared from biomass-based HMF are viable alternatives to the petrochemical benzene-aromatic polyesters, they can serve as low-melting heat bondable fiber, high gas-barrier packaging material, as well as specialty material for engineering applications.

## 1. Introduction

Polyesters are popular synthetic polymers that are closely related to the development of human society. In recent years, there has been significant interest in biomass-based products, which would be alternatives to petroleum-based products, due to their environmental advantages [1]. A recent study predicted that the worldwide capacity of biomass-based polyesters would increase from 0.36 Mt in 2007 to 3.45 Mt in 2020 [2]. It was reported that levulinic acid [3], lactic acid [4], isosorbide [5], succinic acid [6], dodecanedioic acid [7], and ethylene glycol [8] were all potential building blocks for these the preparation of biomass-based polymers. Among them, 2,5-furandicarboxylic acid (FDCA), an aromatic product in nature, has been considered a “sleeping giant” and suitable replacement for terphthalic acid (TPA) in engineering plastics due to similarities between TPA and FDCA [9,10].

FDCA-based polyesters were first studied by Moore and co-workers [11]. After that, Gandini and co-workers [12,13] synthesized some FDCA-based polyesters with interfacial polycondensations and melt polytransesterification and revealed that the mechanical properties, thermal properties, and crystal structures of the obtained poly(butylene-FDCA) (PBF) were similar to petroleum derived poly(butylene terephthalate) (PBT). Based on these works, Gomes [14] and Ma [15] synthesized a series of FDCA-based polyesters with variety of diols, and provided ample evidence in favor of the exploitation of furan monomers as renewable alternatives to petroleum-based aromatic homologs. Subsequently, Sousa [5], Shirke [16], and Pellis [17,18] further synthesized a series of fully renewable poly((ether)ester)s from FDCA and revealed that the obtained polyesters showed better thermal properties than their petroleum-based counterparts. Papageorgiou [19] highlighted the progress and fundamental aspects for the synthesis of bio-based 2,5-FDCA polyesters and their thermal properties, they associated with the coloration and successful syntheses of polyesters with high molecular weights are thoroughly discussed. Recently, Wang and co-workers [20] modified PEF with *trans*- or *cis*-1,4-cyclohexanedimethanol (CHDM) and obtained a polyester with high crystalline, melting temperature, and air barrier property. The above researches have provided important clues that the FDCA-based polyesters can be an ideal substitute of fossil-based counterparts. To fit well with the sustainable conversion process concept and realize the scale production of FDCA-based polyesters, the discovery of new pathway for the production of FDCA is always in demand.

In early reports, homogeneous metal salts, such as Co^2+^/Mn^2+^/Br, have been used in the oxidation of 5-hydroxymethylfrufural (HMF) into FDCA [21,22,23]. However, the use of homogeneous catalysts often suffers from the recycling problem and the toxic pollution of the environment. Currently, plentiful researches reported the use of noble metal catalysts for the oxidation of HMF to FDCA under oxygen atmosphere [24,25,26,27,28,29,30,31,32,33,34,35]. However, high reaction temperature, long reaction time, and high catalysts dosage were always demanded to obtain a high FDCA yield.

Recently, we have developed a method to get purified FDCA from HMF under mild conditions [36,37]. In the present study, we further optimized our reported technical route to get a purified FDCA in high yield. Subsequently, the obtained FDCA, along with commercially available 1,6-hexanediol (HDO), which can be also made from HMF (Scheme 1) [38], were used as the monomers for the synthesis of poly(hexylene 2,5-furandicarboxylate) (PHF). Triphenyl phosphite was used as the antioxidant to alleviate the discoloration problem during the esterification and thus improved the properties of the obtained PHF. Based on this research, we hope that totally biomass-based polyesters with high performance could finally be developed in the future.

## 2. Experimental Section

### 2.1. Materials

HDO was purchased from Xinmingtai Chemical Reagent Co., Ltd. (Wuhan, China). Tetrabutyl titanate was purchased from Kemiou Chemical Reagent Co., Ltd. (Tianjin, China). Phenol, 1,1,2,2-tetrachloroethane, and triphenyl phosphite were purchased from Macklin Biochemical Co., Ltd. (Shanghai, China). HCl (37%) was purchased from Sanying Chemical Reagent Co., Ltd. (Zhejiang, China). HMF was purchased from Wutong Aroma Chemicals Co., Ltd. (Shangdong, China). All of the reagents were analytical grade unless mentioned elsewhere and used as received.

### 2.2. Monomer Synthesis and PHFs Preparation

The oxidation of HMF to FDCA was carried out in a 100 mL high pressure stainless-steel reactor (Anhui Kemi Machinery Technology Co., Ltd., Anhui, China) with the following steps (Scheme 1): (1) 10 mL distilled water and 0.016 mol of NaOH were added into the reactor. (2) The reactor was placed onto a magnetic stirrer and 0.1 mol of HMF, 0.015 mol of prepared K_2_FeO_4_, 0–0.016 mol of K_2_HPO_4_, and 0.005 mmol of metal compound were added into the prepared alkali solution under agitation (400 rpm). (3) The agitation continued for 15 min and the reaction mixture was filtered. (4) The filter residue was dried to get Fe_2_O_3_ so it could be used as the iron source to prepare K_2_FeO_4_. (5) The filtrate was acidized with hydrochloric acid under stirring until a large amount of white precipitate appeared. (6) The precipitate was filtered and the filter residue was dried by vacuum drying for 24 h to obtain FDCA, the filtered water could be reused in another HMF oxidation reaction.

PHFs were synthesized via the direct esterification method, which was performed, as follows: (1) a mixture of HDO (0.01–0.03 mol), FDCA (0.01 mol), tetrabutyl titanate (0.03 mmol) and triphenyl phosphite (0.05–0.5 mmol) was loaded into a 50 mL three-neck round-bottomed flask, which was sealed and purged with N_2_ three times. (2) The flask was then heated to 180 °C with stirring at 120 rmp until the reaction system reached a clear point, and no liquid precipitation was present in the condenser tube. (3) The system pressure was decreased to 600 Pa via vacuum force technology, the temperature was increased to 230–250 °C to start the polycondensation. (4) After polycondensation, the product was dissolved in phenol-tetrachloroethane, then precipitated in methanol three times at the end of the polycondensation, the final PHFs were obtained via vacuum drying at 50 °C for 24 h.

In a typical approach for the preparation of PHF membrane, 0.7 g PHF was dissolved in 10 mL 1,1,2,2-tetrachloroethane to get the PHF solution. Subsequently, the obtained PHF solution was transferred to a 10 mL plastic syringe with an 18-gauge blunt tip needle. For the electrospinning process, a high voltage of 15 kV and a flow rate of 0.001 mm/s were applied, with a distance of 20 cm between the needle and the rotating grounded collector.

### 2.3. Techniques

The intrinsic viscosity (η) of PHFs was measured at a concentration of 0.5 to 1.5 g/dL in 1,1,2,2-tetrachloroethane/phenol (1:1 *w*/*w*) under 25 °C by using a Koehler viscometer and related standard method [39]. The intrinsic viscosity was calculated using the following equations:
(1)$$\left[\eta \right]=\frac{\sqrt{1+1.4\eta_{\mathrm{sp}}}-1}{0.7c}$$
where η_sp_ represented specific viscosity and c was the concentration of PHFs in 1,1,2,2-tetrachloroethane/phenol (1:1 *w*/*w*) at 25 °C.

The mechanical properties of PHF membranes were tested using the universal tensile tester, which were performed on an INSTRON-1121 tester with a strain rate of 5 mm/min at room temperature. Three rectangular specimens (15 mm × 3.23 mm × 3.20 μm) were employed for each test to determine the average of tensile modulus (*E*), tensile strength (σ_m_), and elongation at break (ε_b_). The length and width of the film were measured by Vernier Caliper, and the thickness was measured by a spiral micrometer. The average value was obtained from five times of data. The morphology of PHF membranes fracture surface after membrane stretching was characterized by S4800 SEM (Hitachi, Tokyo, Japan) at the accelerating voltage of 15.00 kV by stretching both ends of the length.

The ^1^H Nuclear Magnetic Resonance (NMR) and ^13^C NMR measurements were carried out on a FTNMR Digital NMR spectrometer (Bruker, Karlsruhe, Germany) operating at 399.95 MHz for ^1^H and 100.58 MHz for ^13^C at room temperature with a magnetic field of 9.4 T. The acquisition time was 0.034 s, the delay time was 2 s, and the proton 90° pulse time was 4.85 s. The PHF sample was dissolved with CF_3_COOD with tetramethylsilane (TMS) as the internal reference.

The Fourier-transform Infrared (FTIR) data of PHF was obtained from FTIR spectrometer (SpectrumOne, Thermo Electron Corporation, Waltham, MA, USA) with a scan of 32 times and a resolution of 4 cm^−1^ in the range of 3500–500 cm^−1^, for which the PHF was pelletized with KBr.

The X-ray diffraction (XRD) patterns of PHFs were recorded on a D8 advance diffractometer (Bruker, Karlsruhe, Germany) with a Cu Kα radiation (λ = 0.154 nm) at 40 KV and 30 mA. PHF was scanned in the 2θ range of 10–35° at a scan rate of 10 °/min.

Differential Scanning Calorimeter (DSC) measurements of PHFs were performed on a differential scanning calorimeter (TA Instruments, New Castle, DE, USA). Measurements were performed under nitrogen atmosphere with the flow rate of 50 mL/min. About 6 mg of PHF was heated to 250 °C at a heating rate of 5 °C/min and then held at this temperature for 3 min in order to erase thermal history. Afterwards, it was cooled down to room temperature at a rate of 5 °C/min and subsequently heated to 250 °C with the same heating rate for the second time. The PHF sample was quenched in liquid nitrogen. 6 ± 0.1 mg sample was used in the test. The sample was sealed in aluminum pans and heated to 250 °C at a heating rate of 5 °C/min.

The thermal stability of PHFs was determined by PYRIS 1 TGA (Perkin-Elmer, Waltham, MA, USA). The thermal analyzer was temperature calibrated using the Curie point of nickel as a reference. The samples of 6 ± 0.5 mg were heated from 20 to 500 °C at a heating rate of 10 °C/min in nitrogen.

The degradability of PHF membranes was evaluated under strong acidic conditions. For a typical procedure, the PHF membranes were made into 1 cm × 1 cm, then the samples were placed in the 50 mL lid bottle. Subsequently, 10 mL of concentrated hydrochloric acid was added into the lid bottle and sealed, the bottle was oscillated at 180 rpm/min frequency for one to four weeks under constant temperature at 25 °C. The solution needed to be replaced with fresh concentrated hydrochloric acid every week. Finally, the treated samples were washed with water, dried at room temperature for 48 h, then weighed. The morphology of PHF membranes after acid treatment was characterized by S4800 SEM (Hitachi, Tokyo, Japan) at the accelerating voltage of 15.00 kV.

## 3. Results and Discussion

### 3.1. FDCA Synthesis

Recently, we have developed a method the oxidation of HMF to FDCA, an 87.2% yield of FDCA was obtained under optimal reaction conditions [36]. It was reported that dipotassium phosphate buffer and metal compounds were helpful in improving the stability of potassium ferrate [40]. To further improve the FDCA yield, we constructed a H_2_O-K_2_HPO_4_-NaOH reaction system with K_2_FeO_4_ as the oxidant and metal compounds, such as NaCl, KCl, CaCl_2_, Mg(OH)_2_, Al(OH)_3_, MnO_2_, Fe_2_O_3_, Fe(OH)_3_, and CuO as the stabilizer of K_2_FeO_4_, the oxidation sketch and experiment results are shown in Figure 1. As can be seen from Figure 1a, the filter residue after the oxidation can be recycled to prepare K_2_FeO_4_, the filtered water can be neutralized with ammonia and reused in another HMF oxidation reaction.

From Figure 1b, we can find that the FDCA yield gradually increased with the increase of K_2_HPO_4_. However, when the amount of K_2_HPO_4_ was over a critical value and further increased, the yield of FDCA distinctly decreased. The main reason is that PO_4_^3−^ in the solution has a stabilizing effect on K_2_FeO_4_, which will prevent its ineffective decomposition [37]. Therefore, the ineffective decomposition of K_2_FeO_4_ gradually decreased with the addition of K_2_HPO_4_, the yield of FDCA gradually increased accordingly. However, when the amount of K_2_HPO_4_ reached a certain level (0.4 mol/L), PO_4_^3−^ in the system had an inhibitory effect on the oxidation HMF to FDCA, then resulting in lower FDCA yield [41].

It can be also observed from Figure 1c that the addition of NaCl, KCl, CaCl_2_, Mg(OH)_2_, Al(OH)_3_, MnO_2_, and CuO had obvious inhibitory effects on the oxidation of HMF to FDCA. However, adding Fe_2_O_3_ and Fe(OH)_3_ could improve the FDCA yield to some extent, as seen when the FDCA yield increased from 82.5 wt % (control sample) to 85.7 and 86.1 wt %, respectively, when 0.5 mmol/L of Fe_2_O_3_ or Fe(OH)_3_ were added. We then further studied the effect of Fe(OH)_3_ amount on the oxidation of HMF to FDCA, the results are shown in Figure 1d. It shows that the FDCA yield gradually increased with the increase of Fe(OH)_3_, a highest FDCA yield of 91.7 wt % was obtained by adding 1 mmol/L Fe(OH)_3_. However, when the Fe(OH)_3_ concentration was over 1 mmol/L and further increased, the yield of FDCA was slightly decreased. It was reported that Fe(OH)_3_ has a stable effect on K_2_FeO_4_, therefore, adding an appropriate amount of Fe(OH)_3_ was helpful in enhancing the FDCA yield [40]. However, Fe(OH)_3_ has a flocculation function and an excessive amount of Fe(OH)_3_ will affect the reaction of HMF and the precipitation of FDCA, thus reducing the FDCA yield.

It can be observed from the above results that the use of dipotassium phosphate buffer and Fe(OH)_3_ improved the stability of potassium ferrate, thus increasing the oxidation efficiency of HMF to FDCA. The highest FDCA yield of 91.7 wt % was obtained under an optimum reaction condition, the obtained FDCA had high purity (>99%) (Appendix A). The present method is more moderate than the reported methods, which can be conducted in atmosphere at room temperature (25 °C), the reaction just needs 15 min to be finished. Most important, the present method is environmentally friendly, all of the Fe ions and water can be recycled, therefore, the present method will not result in the metal and water pollution for the environment, which offers an effective method for the economic and eco-friendly production of FDCA from renewable biomass-based platform chemical, it fits well into the green conversion process concept.

### 3.2. Synthesis of PHFs

In the present work, the effect of the amount of triphenyl phosphite and condensation temperature on the intrinsic viscosity and mechanical properties of the obtained PHFs were studied, and the results are shown in Figure 2. It can be observed that a small amount of triphenyl phosphite could significantly enhance the intrinsic viscosity, which increased from 0.230 dL/g (PHF-1) to 0.780 dL/g (PHF-2) with the adding amount of triphenyl phosphite increased from 0 mmol to 0.05 mmol. However, when the adding amount of triphenyl phosphite was further increased to 0.2 and 0.5 mmol, the intrinsic viscosity was distinctly decreased to 0.720 dL/g (PHF-3) and 0.656 dL/g (PHF-4). This is mainly due to the excessive adding of triphenyl phosphite, resulting in the inhibition of polycondensation [42]. It also can be observed that the color became more and more lighter with the increase of triphenyl phosphite amount, which was mainly due to the inhibiting effect of triphenyl phosphite on the oxidation degradation of the furan ring at high temperature, and thus reduced the discoloration during the synthesis of PHF [42].

Subsequently, the effect of condensation temperature was further evaluated without adding triphenyl phosphite (PHF-1, PHF-5, and PHF-6) (Figure 2a). The results indicated that the intrinsic viscosity gradually increased with the increase of condensation temperature, and a highest intrinsic viscosity (0.736 dL/g) was obtained at the temperature of 250 °C. However, the PHFs would be significantly carbonized when the condensation temperature exceeded 250 °C. In addition, it was found that the intrinsic viscosity increased remarkably from 0.736 dL/g (PHF-6) to 0.803 dL/g (PHF-7) at 250 °C by adding 0.05 mmol triphenyl phosphite, the result further indicated that a small amount of triphenyl phosphite could enhance the intrinsic viscosity of the obtained PHF and thus contribute to the direct esterification.

### 3.3. Structure Characterization of PHFs

PHF-7 displayed a better performance on mechanical property, thermal property, and acid degradability; therefore, its structure was further determined by NMR and FTIR (Figure 3). From ^1^H NMR, we can find that the resonances of C–H peaks on the furan ring, CH_2_ on the ester, and CH_2_ on the carbon chain appeared at 7.30 (H_f1_), 7.23 (H_f2_), 4.37 (H_a_), 1.83 (H_b_), and 1.52 (H_c_) ppm, respectively. The peak integration ratio of the four ^1^H peaks was 1:2:2:2, which was consistent with the calculated values of the molecular structure of PHF. The ^13^C NMR revealed that the resonance peaks associated the furan ring (C_s_/C_f_) appeared at 142.8 and 115.7 ppm. The chemical shifts related to HDO (C_a_, C_b_, and C_c_) appeared at 63.5, 23.9, and 21.0 ppm, respectively. The chemical shift at 156.6 ppm was ascribed to C_x_. The integration ratio of characteristic peak was 1:1:1:1:1:1, which was consistent with the theoretical calculation of PHF.

The FTIR spectrum and its representative resulting peak assignments are shown in Figure 3c. The bands at 770, 818 and 966 cm^−1^ were the out-of-plane bending vibration of =C–H on furan ring; the band at 1038 cm^−1^ was the asymmetric stretching vibration of =C–O on furan ring; the bands at 1570 and 1508 cm^−1^ were characteristic absorption peaks of –C=C on furan ring; the characteristic absorption peaks of –C=O appeared at 2927 and 2861 cm^−1^, and the adsorption peak of =C–H on furan ring appeared at 3120 cm^–1^. These results indicate the existence of furan ring in PHF-7. In addition, the out-of-plane bending vibration of more than six carbon chains of –C–H is observed at 725 cm^−1^. After the polymerization, the strong absorption bands appeared at about 1268 and 1716 cm^−1^ due to the newly formed C–O and C=O in the ester linkage (C−O−C=O); this result confirms the existence of ester in PHF-7 [43].

### 3.4. Crystallinity Properties

The crystallinity of the obtained PHFs was characterized by XRD; the results are shown in Figure 4. We found that the pronounced crystallinity of the solvent treated PHF-6 and PHF-7 membranes was corroborated by the presence of two sharp signals at 17.1° and 24.9° and a less intense diffraction peak at 13.8°. The characteristic diffraction peak at 2θ = 13.8° (d = 6.42Å) could be ascribed to the (110) plane, the peak at 2θ = 17.06° (d = 5.19 Å) could be assigned to the (010) plane, and the diffraction at 2θ = 24.9° (d = 3.58 Å) was attributed to the (111) plane of PHF, respectively. The unit cell of PHF-6 and PHF-7 should be triclinic according to the published paper [2]. These results exhibited excellent agreement with the published literature [43]. However, for PHF-1, only one wide signal at 2θ = 24.5° can be observed, indicating that PHF-1 was a semi-crystalline polyester, and PHF-6 and PHF-7 were crystalline polyesters. The crystallinity of PHF-1, PHF-6, and PHF-7 was 53.3%, 94.3%, and 96.7%, respectively, based on the XRD data.

### 3.5. Mechanical Properties

The mechanical properties of PHF membranes that were evaluated by means of universal tensile testing are shown in Figure 5a, and the related datum are summarized in Figure 5b. It can be clearly observed from the yielding during stress-train tests that the PHF membranes displayed ductile fracture behavior. The SEM diagram of the fracture zone indicated that the PHF membrane showed typical ductile fracture characteristics [44]. Though there has a huge difference in the intrinsic viscosity between PHF-1 (0.230 dL/g) and PHF-6 (0.736 dL/g) due to the difference condensation temperature (Figure 2), they have a similar average Young’s modulus (*E*), maximum tensile strength (σ_m_), and elongation at break (ε_b_) from Figure 5b, which was mainly due to the difference of their crystallinity. From the XRD results shown in Figure 4, we can find that PHF-1 is a semi-crystalline polyester and PHF-6 is a crystalline polyester, which resulted in a similar mechanical property, even though they had a huge difference in the intrinsic viscosity. However, the *E* and σ_m_ obviously increased to 479 and 36.5 MPa from about 450 and 34 MPa with the addition of triphenyl phosphite, though there was a slight decrease in ε_b_. Therefore, the addition of triphenyl phosphite improved the degree of polymerization of PHFs, thus enhancing its mechanical properties, which can be used for the preparation of high elongation fibre.

### 3.6. Thermal Properties

As PHF-1 has the lowest intrinsic viscosity, PHF-6 has the highest intrinsic viscosity without the addition of tripenyl phosphite, and PHF-7 is a contrast sample of PHF-6 with the addition of tripenyl phosphite; therefore, the three samples were chosen for thermal property evaluation, and the results are shown in Figure 6. Figure 6a displayed the DSC trances of PHF-1, 6, and 7, and the glass transition temperature (T_g_) and melting point (T_m_) were listed in Figure 6d. As we can see, the value of T_g_ and T_m_ increased from 48.1 and 143.8 °C to 48.5 and 145.3 °C with the condensation temperature increase from 230 °C to 250 °C. However, the T_g_ and T_m_ had a slight decrease with the adding of triphenyl phosphite. In addition, it also can be observed from Figure 6a that the sample of PHF-6 and PHF-7 had a sharp melting peak, which was due to the high fusion enthalpy resulting from the high intrinsic viscosity, indicating that PHF-6 and PHF-7 had narrow crystal distribution and high crystalline phase [45,46].

Furthermore, the thermal stability of the PHFs was studied and the results are shown in Figure 6b,c, and an about 5% mass loss of PHF-7 at 100 °C should be the carried moisture during the operation. The temperature values of thermal decomposition onset (T_id_) as well as those of the maximum decomposition rate (T_max_) are summarized in Figure 6d. It can be derived that all the three PHFs have similar T_id_ (about 348 °C), which is higher than their T_m_ (about 144 °C). Therefore, they are thermally stable and can be safely processed at a temperature higher than their melting point. Similarly, the increase of intrinsic viscosity caused the decomposition temperature to rise, and PHF-7 had the highest T_max_ (411 °C) than that of PHF-1 and PHF-6 (about 390 °C), indicating that the adding of triphenyl phosphite had a positive effect on the thermal stability of PHF. These results indicated that the adding of triphenyl phosphite not only alleviated yellowing during the synthesis of PHFs, but it also increased their thermal property. Since the obtained PHFs had a similar maximum decomposition rate, a lower thermal decomposition onset and single glass transition temperature than those of the mentioned PEF in the literature [45], this indicates that the obtained PHFs can be used as low melting point polyester.

### 3.7. Degradability in Strong Acid

The degradability of the three PHF membranes was determined by monitoring the weight loss with time under strong acid conditions, and the results are shown in Figure 7. The weight loss of the three PHF membranes is between 6.4–9.4 wt % when treated with strong acid for one week. The weight loss gradually increased with the increase of the treatment time, and the maximum weight loss reached about 20 wt % after four weeks treatment. As has been discussed in the part of Section 3.4, PHF-1 was a semi-crystalline polyester, and PHF-6 and PHF-7 were crystalline polyesters. Their crystallinity was 53.3%, 94.3%, and 96.7%, respectively. As a result, it can be found from Figure 7a that the acid degradation rate of PHF-1 was higher than that of PHF-6 and PHF-7. In addition, we can find that the acid degradation rate of PHF-6 was lower than that of PHF-7, although it has a lower crystallinity than that of PHF-7, this was mainly due to the higher melting temperature of PHF-6 (T_m_ = 145.3 °C, Figure 6) than that of PHF-7 (T_m_ = 144.2 °C, Figure 6). As a result, the weight loss of PHF-1 (23.0 wt %) was higher than that of PHF-7 (20.9 wt %), and PHF-7 was higher that of PHF-6 (19.8 wt %). When compared with the FDCA-based polyesters that were reported in the published literature [47], the weight loss of PHFs was obviously higher than that of the reported results, which only were 1, 2, and 10 wt % in the fourth week and 2, 5, and 28 wt % in twenty-second week, respectively, under neutral, pH = 4.0 and pH = 12.0 conditions. It shows in Figure 7a that the weight loss is not linearly related to acid treatment time, which may be due to the difference in crystalline and amorphous regions [48]. The environmental condition provided in this paper was harsher than that of the reported, but the degradation rate of the polyester was obviously higher, which exhibited better acid degradation performance.

Furthermore, it shows in Figure 7c that the PHF film surface was obviously eroded after four weeks treatment by strong acid when compared with that of the controlled sample (Figure 7b), and the erosion degree of the sample surface was different (Figure 7c), which may be due to the difference between the crystalline region and the amorphous zone [48].

## 4. Conclusions

In the present work, an economic method for the preparation of FDCA derived from HMF was constructed, which used K_2_FeO_4_ as oxidant and K_2_HPO_4_ and Fe(OH)_3_ as the stabilizer of K_2_FeO_4_. The results revealed that the oxidation efficiency of HMF to FDCA was increased, and a 91.7 wt % of FDCA with high purity (>99%) was obtained at 25 °C in 15 min under air atmosphere.

Subsequently, a totally biomass-based polyester, poly(hexylene 2,5-furandicarboxylate) (PHF), was prepared successfully with the directly use of the obtained FDCA and commercially available HDO as monomers. The degree of polymerization of PHFs was improved with the addition of triphenyl phosphite, a crystalline polyester (PHF-7) with typical (110) plane, (010) plane, and (111) plane at 2θ = 13.78° (d = 6.42 Å), 17.06° (d = 5.19 Å), and 24.9° (d = 3.58 Å) was obtained with 0.05 mmol of triphenyl phosphite was added. As a result, the obtained PHF-7 had the highest intrinsic viscosity (0.803 dL/g), average Young’s modulus (479 MPa), and maximum tensile strength (36.5 MPa) than the other PHFs. Its elongation at break was 216%, which was significantly higher than that of PET (90%) and PEF (3%). Furthermore, the high intrinsic viscosity caused the increase in the enthalpy of fusion with respect to as-synthesized PEF, in this case, the melting peak of PHF-7 was particularly sharp, and the T_g_ and T_m_ of PHF-7 were about 48 and 144 °C, which were relatively lower when compared to those of PEF, which can be served as heat bondable fibre with low melting and packaging application with high gas barrier demanding as well as engineering materials.

All of these results support the notion that it is entirely possible to synthesize the furan-aromatic polyesters via the direct esterification method starting from HMF, and the properties of furan-aromatic polyesters based on renewable resources are favorable to those from the petrochemical benzene-aromatic polyesters.

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
