# Peer review of "An Eco-Friendly Method to Get a Bio-Based Dicarboxylic Acid Monomer 2,5-Furandicarboxylic Acid and Its Application in the Synthesis of Poly(hexylene 2,5-furandicarboxylate) (PHF)"

_polymers, 2019, doi:10.3390/polym11020197_

Round 1

Reviewer 1 Report

Zhang and co-workers present a work on the environmentally friendly synthesis of FDCA and the subsequent synthesis of poly(hexylene 2,5-furandicarboxylate). Despite the work is interesting, I think that the authors should address the points listed below before the manuscript is acceptable for publication.

-        Pag. 2, Line 44. Ethylene diol? Do the authors mean ethylene glycol here?

-        In the introduction the authors should add a comment on the recent works that show how PEF is easier to hydrolyze than PET using cutinases. That’s also a massive advantage of the FDCA-based materials that is missing in the introductory section of this manuscript.

-        I don’t understand what’s the main difference/advancement from the previously published work regarding the first part of the paper, leading to the FDCA production. What’s the novelty of section 3.1 of the present paper vs ref. 16?

-        Did the authors measure the molecular weight and the dispersity of the synthesized polymer? I couldn’t find these data in the manuscript.

-        Why did the authors try the degradation of the polymer only in acid and not in base? It is well known that base is better for hydrolysing polyesters.

-        Please revise the English throughout the manuscript. Some parts are really difficult to understand at the moment.

Author Response

Response to Reviewer 1 Comments

Point 1: Pag. 2, Line 44. Ethylene diol? Do the authors mean ethylene glycol here?

Response 1: Thanks for your kindly comment. “ethylene diol has been changed to “ethylene glycol.

Point 2: In the introduction the authors should add a comment on the recent works that show how PEF is easier to hydrolyze than PET using cutinases. That’s also a massive advantage of the FDCA-based materials that is missing in the introductory section of this manuscript.

Response 2: Thank you for your suggestion. We have added more information in the revised paper (Lines 52-54, 82-85).

Point 3: I don’t understand what’s the main difference/advancement from the previously published work regarding the first part of the paper, leading to the FDCA production. What’s the novelty of section 3.1 of the present paper vs ref. 16?

Response 3: Thanks for your kindly comment. In the present work, a FDCA yield up to 91% could be obtained in atmosphere, and the reaction just need to be conducted at room temperature (25 °C) in 15 min. Most important, the FDCA with a purity up to 99% could easy be obtained and could be directly used in the preparation of PHFs.

Point 4: Did the authors measure the molecular weight and the dispersity of the synthesized polymer? I couldn’t find these data in the manuscript.

Response 4: Thanks for your kindly comment. We did not measure the molecular weight and the dispersity of the synthesized polymer. Because the obtained PHFs were insoluble in common solvents and we could not find a GPC to measure. However, in the present work, we measured the intrinsic viscosity of the obtained PHFs, which could also reflect the molecular weight in some extent.

Point 5: Why did the authors try the degradation of the polymer only in acid and not in base? It is well known that base is better for hydrolysing polyesters.

Response 5: Thanks for your kindly suggestion. FDCA-based polyesters have certain biodegradability, our initial goal was to understand its potential for biodegradation, so we tested the degradation in acid, because the acidic environment was closer to biological environments.

Reviewer 2 Report

The paperAn eco-friendly method to get a bio-based dicarboxylic acid monomer 2,5-furandicarboxylic acid and the synthesis of poly(hexylene 2,5-furandicarboxylate) with the modification of triphenyl phosphitesubmitted for review in Polymers deals mainly about the synthesis of 2,5-furandicarboxylic acid and the next step – synthesis of poly(hexylene 2,5-furandicarboxylate)s from the obtained diacid and 1,6-hexanodiol. Authors claim that the synthesis of diacid is “eco-friendly” and the synthesized polymer can be served as the eco-packaging application.

The article is written chaotically with many errors and inaccuracies. Many of the statements used in the article are incorrect. Some of them are contrary to the knowledge of polymers and physical chemistry of polymers. Some of the cited literature is no related to the described topic (ref. 10).

The most serious errors are:

-        Line 50; “methyl alcohol as the monomer”

-        Line 228; “inhibit the discoloration” means that the color of the sample is more intense with the use of the triphenyl phosphite;

-        Line 267; what is the “molecular structure theory of PHF”?

-        Line 272: what is the “absorption peak” in the NMR technic?

-        Line 281; what is the “furan nucleus of PHF-7”?

-        Line 282; what is the “surface vibration” in FTIR analysis?

-        Line 283; where is the polyglycol segments (visible in the FTIR spectrum) in the synthesized and analyzed copolyester?

-        Line 312-313; the backbone stiffness is not depended on the chain length;

-        Line 323; the increase of intrinsic viscosity cannot rise the temperature of decomposition;

and many others.

In addition, the English language used in the article is incorrect and often incorrectly describing the concept of the paper.

Authors claim that synthesized polymers can be served as the eco-packaging application but it contains a triphenyl phosphite which is “Hazardous to the aquatic environment” and sensitizing and irritating eyes and skin.

Authors claim that the water after the synthesis of diacid can be reuse in the next process but what is with the soluble phosphates?

Authors discussing the degradability of obtained polymers, which is clearly depend on the crystallinity of the sample but it is not estimated.

In conclusion, the paper has to be completely rewritten and some experiments (degree of crystallinity) should be performed. The Abstract, Experimental Part and Result and discussion should be written comprehensibly and succinctly, without repetition.

The terms “eco-frendly” and “bio-based” etc. should not be overused.

Author Response

Response to Reviewer 2 Comments

Point 1: The article is written chaotically with many errors and inaccuracies. Many of the statements used in the article are incorrect. Some of them are contrary to the knowledge of polymers and physical chemistry of polymers. Some of the cited literature is no related to the described topic (ref. 10).

Response 1:  Thanks for your kindly comments.  Ref. 10 has been deleted in the revised paper.

Point 2: Line 50; “methyl alcohol as the monomer”

Response 2: Thank you for your suggestion. The “methyl alcohol as the monomer” has been replaced by “methyl alcohol and diol as the material”.

Point 3: Line 228; “inhibit the discoloration” means that the color of the sample is more intense with the use of the triphenyl phosphite;

Response 3:  Thank you for your suggestion. The inhibit the discoloration has been replaced by reduce the discoloration.

Point 4: Line 267; what is the “molecular structure theory of PHF”?

Response 4: Thank you for your suggestion. The “molecular structure theory of PHF” has been changed into “molecular structure of PHF”.

Point 5: Line 272: what is the “absorption peak” in the NMR technic?

Response 5: Thank you for your suggestion. The “absorption peak” has been changed into “peak”.

Point 6: Line 281; what is the “furan nucleus of PHF-7”?

Response 6: Thank you for your suggestion. The “furan nucleus in PHF-7” has been changed into “furan ring of PHF-7”.

Point 7: Line 282; what is the “surface vibration” in FTIR analysis?

Response 7: Thank you for your suggestion. The “surface vibration” has been replaced by “out-of-plane bending vibration”.

Point 8: Line 283; where is the polyglycol segments (visible in the FTIR spectrum) in the synthesized and analyzed copolyester?

Response 8: Thank you for your suggestion. The “the existence of polyglycol” has been replaced by “that hexanediol”.

Point 9: Line 312-313; the backbone stiffness is not depended on the chain length;

Response 9: Thank you for your suggestion. Revisions have been made accordingly.

The transition temperatures that are observed for the synthesized sample is nothing more than a consequence of its molecular weight.

Point 10: Line 323; the increase of intrinsic viscosity cannot rise the temperature of decomposition;

Response 10: Thank you for your suggestion. The “intrinsic viscosity” has been replaced by “molecular weight”.

Point 11: The English language used in the article is incorrect and often incorrectly describing the concept of the paper. Authors claim that synthesized polymers can be served as the eco-packaging application but it contains a triphenyl phosphite which is “Hazardous to the aquatic environment” and sensitizing and irritating eyes and skin.

Response 11: Thank you for your suggestion. Revisions have been made accordingly.

These totally biomass-based polyesters can be viable alternatives to the petrochemical-based benzene-aromatic polyesters, which can be served as heat bondable fibre with low melting and packaging application with high gas barrier demanding as well as engineering materials.

Point 12: The English language used in the article is incorrect and often incorrectly describing the concept of the paper. Authors claim that the water after the synthesis of diacid can be reuse in the next process but what is with the soluble phosphates?

Response 12: Thanks for your kindly comment. The “eco-friendly method” has been changed into “an economic method” in the revised paper.

Point 13: The English language used in the article is incorrect and often incorrectly describing the concept of the paper. Authors discussing the degradability of obtained polymers, which is clearly depend on the crystallinity of the sample but it is not estimated.

Response 13:  Thank you for your suggestion. Revisions have been made accordingly. The following details were added in the revised manuscript:

As shown in Figure 4(d) PHF-6 and PHF-7 have better crystallinity than that of PHF-1, and it was mentioned that the polymer with the crystallinity.”

Point 14: The English language used in the article is incorrect and often incorrectly describing the concept of the paper. In conclusion, the paper has to be completely rewritten and some experiments (degree of crystallinity) should be performed. The Abstract, Experimental Part and Result and discussion should be written comprehensibly and succinctly, without repetition.

Response 14: Thank you for your suggestion. We have revised the whole article. And the Abstract and Conclusion parts have been reorganized.

Line 17-31: 

Abstract: “Recently, we have developed an eco-friendly method for the preparation of a renewable dicarboxylic acid 2,5-furandicarboxylic acid (FDCA) from biomass-based 5-hydroxymethylfrufural (HMF). In the present work, we optimized our reported method, which used phosphate buffer and Fe(OH)3 as buffer solution and pro-oxidant to improve the stability of potassium ferrate, and then got. purified FDCA (purity was up to 99%) in high yield (91.7 wt%) under mild conditions (25 °C, 15 min, air atmosphere). Then, the obtained FDCA, along with 1,6-hexanediol (HDO), which was also made from HMF, were used as monomers for the synthesis of a totally biomass-based polyester, poly(hexylene 2,5-furandicarboxylate) (PHF) via direct esterification with the modification of triphenyl phosphite. And the mechanical properties, molecular structure, thermal properties, and degradability of the obtained PHFs was measured or characterized by Koehler viscometer, XRD, FTIR, NMR, DSC, DTG, SEM, and weighting method. The experimental evidence clearly showed that the furan-aromatic polyesters prepared from biomass-based HMF are viable alternatives to the petrochemical benzene-aromatic polyesters, and they can serve as heat bondable fibre with low melting and packaging application with high gas barrier demanding as well as engineering materials.”

Line 358-380: 

Conclusion: "In the present work, an economic method for the preparation of FDCA derived from HMF was constructed, which used K2FeO4 as oxidant, K2HPO4 and Fe(OH)3 as the stabilizer of K2FeO4. The results revealed that the oxidation efficiency of HMF to FDCA was increased, and a 91.7 wt% of FDCA with high purity (>99%) was obtained at 25 °C in 15 min under air atmosphere.

Then, a totally biomass-based polyester, poly(hexylene 2,5-furandicarboxylate) (PHF), was prepared successfully with the directly use of the obtained FDCA and commercially available HDO as monomers. And the degree of polymerization of PHFs was improved with the addition of triphenyl phosphite, a crystalline polyester (PHF-7) with typical (110) plane, (010) plane, and (111) plane at 2θ=13.78° (d=6.42Å), 17.06° (d=5.19Å), and 24.9° (d=3.58 Å) was obtained with 0.05 mmol of triphenyl phosphite was added. As a result, the obtained PHF-7 had the highest intrinsic viscosity (0.803 dL/g), average Young’s modulus (479 MPa) and maximum tensile strength (36.5 MPa) than the other PHFs. And its elongation at break was 216%, which was significantly higher than that of PET (90%) and PEF (3%). Furthermore, the high intrinsic viscosity caused the increase in the enthalpy of fusion with respect to as-synthesized PEF, in this case, the melting peak of PHF-7 was particularly sharp, and the single glass transition temperature (Tg) and melting temperature (Tm) of PHF-7 were about 48 °C and 144 °C, which were relatively lower compared to those of PEF, which can be served as heat bondable fibre with low melting and packaging application with high gas barrier demanding as well as engineering materials.

All these results support the notion that it is entirely possible to synthesize the furan-aromatic polyesters via direct esterification method starting from HMF, and the properties of furan-aromatic polyesters based on renewable resources are favorable to those from the petrochemical benzene-aromatic polyesters.”

Point 15: The English language used in the article is incorrect and often incorrectly describing the concept of the paper. The terms “eco-friendly” and “bio-based” etc. should not be overused

Response 15: Thank you for your suggestion. We have carefully revised our paper and standardized our terminology. And the terms of “eco-friendly” and “bio-based” have been reduced in the revised paper.

Round 2

Reviewer 1 Report

Together with the work from Shirke and coworkers, the authors sshould cite also the two papers from the Guebitz group regarding the PEF vs PET degradation topic in the introduction (first works on the topic). With this add to the intro section I find the manuscript suitable for publication. 

Author Response

Response to Reviewer 1 Comments

Point 1: Together with the work from Shirke and coworkers, the authors should cite also the two papers from the Guebitz group regarding the PEF vs PET degradation topic in the introduction (first works on the topic). With this add to the intro section I find the manuscript suitable for publication.

Response 1: Thank you for your kindly suggestion. We have added the two references in the revised paper.

Line 51-53: “Then, Sousa5, Shirke14 and Pellis15-16 further synthesized a series of fully renewable poly((ether)ester)s from FDCA and revealed that the obtained polyesters showed better thermal properties than their fossil-based counterparts.”

Reviewer 2 Report

The paperAn eco-friendly method to get a bio-based dicarboxylic acid monomer 2,5-furandicarboxylic acid and the synthesis of poly(hexylene 2,5-furandicarboxylate) with the modification of triphenyl phosphite” resubmitted for review in Polymers is still written chaotically with many errors and inaccuracies. Some amendments have been removed, though many have been removed incorrectly.

First of all, authors claim in the Abstract and Conclusions parts that the substrates are from bio-based HMF but in the Experimental part authors state that the HMF was purchased from Wutong Aroma Chemicals Co. Ltd. Is the HMF really from biomass?

Authors use in many places the phrase “modification of triphenyl phosphite” – authors should clearly described whether the modification concerns the method or PHF and change the paper title, which is now misleading.

The introduction is long but there is not mentioned the other methods for the synthesis of FDCA (described also in patents) and also PHFs. Some references are in this part are discussed too wide. All that part should be totally rewritten.

The experimental part should be written in such a way that the experiment can be repeated. Some sentences are poorly constructed (see line 122-123).

The part 3. Result and discussion is written very chaotically. The influence of catalyst has to be clearly presented. Also the influence of the triphenyl phosphite addition and the reaction temperature have to be discussed more clear.

In line 272 – is really hydroxyl carbon on the PHF backbone? The same is in the line 285. All the discussion of the NMR spectra (1H and 13C) is too long.

There are a lot of strange sentences as in line 307-308, 324-325 etc. The corrected sentence in line 314 and the reference (20) is not related to the observed relationship.

Authors should clearly described the influence of the temperature of the PHF synthesis on the molecular mass of obtained polymers and also their Tg and Tm.

The crystallinity of synthesized polymers should be estimated (not as better or less) since it is the key to the discussion of the stability and mechanical properties.

Above, there are only some major remarks but all paper should be carefully corrected.

In conclusion, the paper should be completely rewritten and corrected. The language has to be corrected by the native speaker. In such form the paper cannot be published in Polymers.

Author Response

Response to Reviewer 2 Comments

Point 1: First of all, authors claim in the Abstract and Conclusions parts that the substrates are from bio-based HMF but in the Experimental part authors state that the HMF was purchased from Wutong Aroma Chemicals Co. Ltd. Is the HMF really from biomass?

Response 1:  Thanks for your kindly comments. Actually, HMF has been described as a “Top 10” chemical from carbohydrates by US Department of Energy (DOE) in 2004. And as far as we know, glucose was used as the raw material to produce HMF in Wutong Aroma Chemicals Co. Ltd, which can be derived from biomass. Therefore, we used “bio-based HMF” in this paper.

Point 2: Authors use in many places the phrase “modification of triphenyl phosphite” – authors should clearly described whether the modification concerns the method or PHF and change the paper title, which is now misleading.

Response 2: Thank you for your kindly suggestion. The title has been changed into “An Eco-friendly Method to Get a Bio-based Dicarboxylic acid Monomer 2,5-Furandicarboxylic Acid and its application in the Synthesis of Poly(hexylene 2,5-furandicarboxylate) (PHF)”.

In addition, the sentence of “Then, the obtained FDCA, along with 1,6-hexanediol (HDO), which was also made from HMF, were used as monomers for the synthesis of a totally biomass-based polyester, poly(hexylene 2,5-furandicarboxylate) (PHF) via direct esterification with the modification of triphenyl phosphite” has been changed into “Then, the obtained FDCA, along with 1,6-hexanediol (HDO), which was also made from HMF, were used as monomers for the synthesis of a totally biomass-based polyester, poly(hexylene 2,5-furandicarboxylate) (PHF) via direct esterification, and triphenyl phosphite was used as the antioxidant to alleviate the discoloration problem during the esterification” (Line 20-24).

Point 3: The introduction is long but there is not mentioned the other methods for the synthesis of FDCA (described also in patents) and also PHFs. Some references are in this part are discussed too wide. All that part should be totally rewritten.

Response 3:  Thank you for your kindly suggestion. The “Introduction” has been rewritten in the revised paper.

Line 45-67: “FDCA-based polyesters were first studied by Moore and co-workers.9 After that, Gandini and co-workers10-11 synthesized some FDCA-based polyesters with interfacial polycondensations and melt polytransesterification and revealed that the mechanical properties, thermal properties and crystal structures of the obtained PBFs were similar to petroleum derived PBT. Based on these works, Gomes12 and Ma13 synthesized a series of FDCA-based polyesters with variety of diols, and provided ample evidence in favor of the exploitation of furan monomers as renewable alternatives to fossil-based aromatic homologs. Then, Sousa5, Shirke14 and Pellis15-16 further synthesized a series of fully renewable poly((ether)ester)s from FDCA and revealed that the obtained polyesters showed better thermal properties than their fossil-based counterparts. And Papageorgiou17 highlighted the progress and fundamental aspects for the synthesis of bio-based 2,5-FDCA polyesters and their thermal properties, and associated with the coloration and successful syntheses of polyesters with high molecular weights are thoroughly discussed. Recently, Wang and co-workers18 modified PEF with trans- or cis-1,4-cyclohexanedimethanol (CHDM) and obtained a polyester with high crystalline, melting temperature, and air barrier property. The above researches have provided important clues that the FDCA-based polyesters can be an ideal substitute of fossil-based counterparts. To fit well with the sustainable conversion process concept and realize the scale production of FDCA-based polyesters, the discovery of new pathway for the production of FDCA is always in demand.

In early reports, homogeneous metal salts, such as Co2+/Mn2+/Br, have been used in the oxidation of HMF into FDCA.19-21 However, the use of homogeneous catalysts often suffer from the recycling problem and the toxic pollution of the environment. Currently, plentiful researches reported the use of noble metal catalysts for the oxidation of HMF to FDCA under oxygen atmosphere.22-31 However, high reaction temperature, long reaction time and high catalysts dosage were always demanded to obtain a high FDCA yield.”

Point 4: The experimental part should be written in such a way that the experiment can be repeated. Some sentences are poorly constructed (see line 122-123).

Response 4: Thank you for your kindly suggestion. This part has been rewritten in the revised paper.

Line 97-105: “PHFs were synthesized via direct esterification method, which was performed as follows: 1) a mixture of HDO, FDCA, tetrabutyl titanate and triphenyl phosphite was loaded into a 50 mL three-neck round-bottomed flask, which was sealed and purged with N2 three times. 2) The flask was then heated to 180 °C with stirring at 120 rmp until the reaction system reached a clear point, and no liquid precipitation was present in the condenser tube. 3) The system pressure was decreased to 600 Pa via vacuum force technology, and the temperature was increased to 230-250 °C to start the polycondensation. 4) After the polycondensation, the product was dissolved in phenol-tetrachloroethane and then precipitated in methanol three times at the end of the polycondensation, and the final PHFs were obtained via vacuum drying at 50 °C for 24 h.”

Point 5: The part 3. Result and discussion is written very chaotically. The influence of catalyst has to be clearly presented. Also the influence of the triphenyl phosphite addition and the reaction temperature have to be discussed more clear.

Response 5: Thank you for your kindly suggestion. The influence of the triphenyl phosphite addition and the reaction temperature has been discussed in the revised paper.

Line 196-207: “In the present work, the effect of the amount of triphenyl phosphite and condensation temperature on the intrinsic viscosity and mechanical properties of the obtained PHFs were studied, and the results are shown in Figure 2. It can be observed from Figure 2(a) (PHF-1 and PHF-2) that a small amount of triphenyl phosphite could significantly enhance the intrinsic viscosity, which increased from 0.230 dL/g to 0.780 dL/g with the adding amount of triphenyl phosphite increased from 0 mmol to 0.05 mmol. However, when the adding amount of triphenyl phosphite was further increased to 0.2 mmol and 0.5 mmol, the intrinsic viscosity was distinctly decreased to 0.720 dL/g (PHF-3) and 0.656 dL/g (PHF-4). This is mainly due to the excessive adding of triphenyl phosphite resulting in the inhibition of polycondensation.38 In addition, it also can be observed that the color became more and more lighter with the increase of triphenyl phosphite amount. This was mainly due to the inhibiting effect of triphenyl phosphite on the oxidation degradation of furan ring at high temperature, and thus reduced the discoloration during the synthesis of PHF.38

Line 211-219: “Furthermore, the effect of condensation temperature was further evaluated without adding triphenyl phosphite (PHF-1, PHF-5, and PHF-6) (Figure 2(a)). The results indicated that the intrinsic viscosity gradually increased with the increase of condensation temperature, and a highest intrinsic viscosity (0.736 dL/g) was obtained at the temperature of 250 . However, the PHFs would be significantly carbonized when the condensation temperature exceeded 250 . In addition, we found that the intrinsic viscosity increased remarkably from 0.736 dL/g (PHF-6) to 0.803 dL/g (PHF-7) at 250 by adding 0.05 mmol triphenyl phosphite, the result further indicated that a small amount of triphenyl phosphite could enhance the intrinsic viscosity of the obtained PHF and thus contribute to the direct esterification.”

Point 6: In line 272 – is really hydroxyl carbon on the PHF backbone? The same is in the line 285. All the discussion of the NMR spectra (1H and 13C) is too long.

Response 6: Thank you for your kindly comment. This part has been rewritten in the revised paper.

Line 239-247: “From 1H NMR we can find that the resonance of C-H peaks on the furan ring, CH2 on the ester, and CH2 on the carbon chain appeared at 7.30 (Hf1), 7.23 (Hf2), 4.37 (Ha), 1.83 (Hb), and 1.52 (Hc) ppm, respectively. The peak integration ratio of the four 1H peaks was 1:2:2:2, which was consistent with the calculated values of the molecular structure of PHF. The 13C NMR revealed that the resonance peaks associated the furan ring (Cs/Cf) appeared at 142.8 and 115.7 ppm. The chemical shifts related to HDO (Ca, Cb, and Cc) appeared at 63.5, 23.9 and 21.0 ppm, respectively. And the chemical shift at 156.6 ppm was ascribed to Cx. The integration ratio of characteristic peak was 1:1:1:1:1:1, which was consistent with the theoretical calculation of PHF.”

Point 7: There are a lot of strange sentences as in line 307-308, 324-325 etc. The corrected sentence in line 314 and the reference (20) is not related to the observed relationship.

Point 8: Authors should clearly described the influence of the temperature of the PHF synthesis on the molecular mass of obtained polymers and also their Tg and Tm.

Response 7&8: Thank you for your kindly advise. The two pars have been rewritten in the revised paper.

Line 279-286: “Figure 5(a) displayed the DSC trances of PHF-1, 6, and 7, and the glass transition temperature (Tg) and melting point (Tm) were listed in Figure 5(d). As we can see, the value of Tg and Tm increased from 48.1 and 143.8 °C to 48.5 and 145.3 °C with the condensation temperature increased from 230 °C to 250 °C. However, the Tg and Tm had a slight decrease with the adding of triphenyl phosphite. In addition, it also can be observed from Figure 5(a) that the sample of PHF-6 and PHF-7 had a sharp melting peak, which was due to the high fusion enthalpy resulted by the high intrinsic viscosity, indicating PHF-6 and PHF-7 had narrow crystal distribution and high crystalline phase.41-42

Line 292-294: “Similarly, the increase of intrinsic viscosity caused the decomposition temperature to rise, and PHF-7 had the highest Tmax (411 °C) than that of PHF-1 and PHF-6 (about 390 °C), indicating that the adding of triphenyl phosphite had a positive effect on the thermal stability of PHF.”

Point 9: The crystallinity of synthesized polymers should be estimated (not as better or less) since it is the key to the discussion of the stability and mechanical properties.

Response 9: Thank you for your kindly advise. And the crystallinity of PHFs has been estimated in “3.5. Crystallinity properties”.

Line 262-272: “The crystallinity of the obtained PHFs was characterized by XRD, the results are shown in Figure 4. We found that the pronounced crystallinity of the solvent treated PHF-6 and PHF-7 membranes was corroborated by the presence of two sharp signals at 17.1° and 24.9° and a less intense diffraction peak at 13.8°. The characteristic diffraction peak at 2θ=13.8° (d=6.42Å) could be ascribed to the (110) plane, the peak at 2θ=17.06° (d=5.19 Å) could be assigned to the (010) plane, and the diffraction at 2θ=24.9° (d=3.58 Å) was attributed to the (111) plane of PHF, respectively. The unit cell of PHF-6 and PHF-7 should be triclinic according to the published paper.2 These results exhibited excellent agreement with the published literature.35 However, for PHF-1, only one wide signal at 2θ=24.5° can be observed, indicating that PHF-1 was a semi-crystalline polyester, and PHF-6 and PHF-7 were crystalline polyesters. This result is consistent with the phenomenon of high weight loss for PHF-1 during the strong acid treatment.”

Round 3

Reviewer 2 Report

Review of the third version of the paper titled “An Eco-friendly Method to Get a Bio-based Dicarboxylic acid Monomer 2,5-Furandicarboxylic Acid and its application in the Synthesis of Poly(hexylene 2,5-furandicarboxylate)”.

The paper resubmitted for the review has been corrected very carelessly despite these amendments, it contains a lot of inaccuracies, errors and unreliable discussion of results. The last review contained information that only some of the major errors were listed which do not allow the publication of this article. Authors corrected some of listed errors (the most important are corrected carelessly) but several serious mistakes have remained.

First of all, polymer properties, such as degradability, mechanical and thermal properties strongly depend on the sample morphology (amorphous/crystalline, semicrystalline) and the morphology strongly depends on the thermal history of the sample and also depends on the method of the sample preparation. Authors compare mechanical and thermal properties of prepared polymers with the properties of PET but only of the crystalline morphology (εb = 90%). In the same reference, cited by authors, the same parameter for the semicrystalline PET is εb = 250%. In this case the content of crystalline phase of the prepared and tested samples of polymers have to be estimated in %. It can be calculated using the XRD or DSC methods. Also the mechanical measurement have to be clearly described with particular emphasis on sample preparation (electrospinned membrane) (how the average values were calculated, how the sample (membrane) was fixed in the tester, how was the membrane structure (fiber diameter), etc.).

How the authors explain the mechanical properties of PHF1 and PHF6 samples (almost the same) in comparing their huge difference in the measured intrinsic viscosity (0.23 dL/g and 0.72 dL/g respectively)?

The next serious problem and mistake is related to cited literature:

-        There are several cited references which cite another reference. The source literature have to be cited.

-        Several cited literature references are not on the subject (ref. 3, 4, 6, 7, 8 – not described the study of the synthesis of polymers but describe the preparation of such chemicals).

-        References 35, 38, 39, 46 are completely not related to the discussed topic.

-        References placed on the figures are not corrected in accordance with the changes in the text of the paper.

Figure 4 contain too small letters and digits.

The discussion in line 280 – 296 is very complicated especially because the authors mistook the parameter symbols and used some cited data very selectively.

The synthesis of the polymers (PHFs preparation) is not described properly - no quantities of reagents were given.

Line 316-317 - authors observed signals derived from chains of more than six carbon  - is it really such chain in obtained polymers?

Line 331-332 - the degradation has not been discussed yet up to this part.

Line 329-331 - How the authors explain the mechanical properties of PHF1 and PHF6 and PHF7 samples based on the observed crystallinity (PHF6 and PHF7 are crystalline but the Young`s modulus is lower than the semi crystalline PHF1 – another parameters are almost the same).

The discussion on thermal properties of obtained polymers in line 347 – 359 is completely incomprehensible and inconsistent with the data in Figure 5.

-        In fig 5c the PHF1 sample has the highest decomposition temperature but in the fig 5b the temperature is lower.

-        How was the thermal history of samples investigated by the DSC method on the fig 5a (is it the first run or second or after quenching). In the experimental part the preparation is described very precisely but there is no information which run is presented in the fig 5a.

-        The sample PHF7 on the fig. 5b lost about 5% of mass already at a temperature of 250 °C, unlike the samples PHF1 and PHF6. This is not mentioned and not discussed in the text.

-        How consistent are the results of mechanical and thermal tests - were the samples prepared in the same way for mechanical and thermal tests? Data from DSC and XRD tests (if they are true) should be consistent and confirm mechanical properties.

The discussion of the degradability of the obtained polymers in the strong acidic conditions is very intricate. The authors chose an extremely acid environment without giving reasons for their choice. Comparing the degradation of the polymers obtained with the described degradation experiments of other polymers under completely different conditions does not lead to any conclusions. The reference 46 is related to enzymatic degradation and therefore it is not suitable for this discussion.

Line 366 – “weightlessness” means the lack of gravity and not the loss of weight

The ref. 38 is cited not properly since “van” is only part of the surname (van Lith).

In conclusion, the paper still have many inaccuracies, errors and unreliable discussion of results. Discussed and measured parameters are not consistent. The cited literature does not help the reader to understand the paper and sometimes even misleads.

In this case the paper in such form cannot be published in Polymers.

Author Response

Response to Reviewer 2 Comments

Point 1: First of all, polymer properties, such as degradability, mechanical and thermal properties strongly depend on the sample morphology (amorphous/crystalline, semicrystalline) and the morphology strongly depends on the thermal history of the sample and also depends on the method of the sample preparation. Authors compare mechanical and thermal properties of prepared polymers with the properties of PET but only of the crystalline morphology (εb = 90%). In the same reference, cited by authors, the same parameter for the semicrystalline PET is εb = 250%. In this case the content of crystalline phase of the prepared and tested samples of polymers have to be estimated in %. It can be calculated using the XRD or DSC methods. Also the mechanical measurement have to be clearly described with particular emphasis on sample preparation (electrospinned membrane) (how the average values were calculated, how the sample (membrane) was fixed in the tester, how was the membrane structure (fiber diameter), etc.).

Response 1:  Thanks for your kindly suggestion, we have made some improvements in the revised paper according to your advice.  And the crystallinity has been calculated based on the data of XRD.

Line 121-125: The length and width of the film were measured by Vernier Caliper, and the thickness was measured by spiral micrometer. The average value was obtained from five times of data. The morphology of PHF membranes fracture surface after membrane stretching was characterized by S4800 SEM (Hitachi, Japan) at the accelerating voltage of 15.00 kV by stretching both ends of the length.

Line 260-261: the crystallinity of PHF-1, PHF-6, and PHF-7 has been given. “And the crystallinity of PHF-1, PHF-6 and PHF-7 was 53.3%, 94.3% and 96.7%, respectively based on the XRD data.”

Line 264-281: The mechanical properties were set to a separate section and the discussion has been rewritten. “The mechanical properties of PHF membranes which were evaluated by means of universal tensile testing are shown in Figure 5(a), and the related datum are summarized in Figure 5(b). It can be clearly observed from the yielding during stress-train tests that the PHF membranes displayed ductile fracture behavior. And the SEM diagram of the fracture zone indicated that the PHF membrane showed typical ductile fracture characteristics.39 Though there has a huge difference in the intrinsic viscosity between PHF-1 (0.230 dL/g) and PHF-6 (0.736 dL/g) due to the difference condensation temperature (Figure 2), they have a similar average Young’s modulus (E), maximum tensile strength (σm) and elongation at break (εb) from Figure 5(b), which was mainly due to the difference of their crystallinity. From the XRD results show in Figure 4 we can find that PHF-1 is a semi-crystalline polyester and PHF-6 is a crystalline polyester, which resulted in a similar mechanical property even they had a huge difference in the intrinsic viscosity. However, the E and σm obviously increased to 479 MPa and 36.5 MPa from about 450 MPa and 34 MPa with the addition of triphenyl phosphite, though there was a slight decrease in εb. Therefore, the addition of triphenyl phosphite improved the degree of polymerization of PHFs, thus enhancing its mechanical properties, which can be used for the preparation of high elongation fibre.

Figure 5. The mechanical properties of PHF membranes.

Point 2: How the authors explain the mechanical properties of PHF1 and PHF6 samples (almost the same) in comparing their huge difference in the measured intrinsic viscosity (0.23 dL/g and 0.72 dL/g respectively)?

Response 2: Thank you for your comment. We have given some explain in the revised paper.

Line 269-275: “Though there has a huge difference in the intrinsic viscosity between PHF-1 (0.230 dL/g) and PHF-6 (0.736 dL/g) due to the difference condensation temperature (Figure 2), they have a similar average Young’s modulus (E), maximum tensile strength (σm) and elongation at break (εb) from Figure 5(b), which was mainly due to the difference of their crystallinity. From the XRD results show in Figure 4 we can find that PHF-1 is a semi-crystalline polyester and PHF-6 is a crystalline polyester, which resulted in a similar mechanical property even they had a huge difference in the intrinsic viscosity.”

Point 3: There are several cited references which cite another reference. The source literature has to be cited.

Response 3: Thank you for your comment. Ref. 9, 10, 34, 35 have been cited.

Point 4: Several cited literature references are not on the subject (ref. 3, 4, 6, 7, 8 – not described the study of the synthesis of polymers but describe the preparation of such chemicals).

Response 4: Thank you for your suggestion. This sentence has been rewritten in the revised paper.

Line 39-41: And it was reported that levulinic acid,3 lactic acid,4 isosorbide,5 succinic acid,6 dodecanedioic acid,7 and ethylene glycol 8 were all potential building blocks for these the preparation of biomass-based polymers.

Point 5: References 35, 38, 39, 46 are completely not related to the discussed topic.

Response 5: Thanks for your kindly suggestion. We have confirmed the references and revised it.

Point 6: References placed on the figures are not corrected in accordance with the changes in the text of the paper.

Response 6: Thanks for your kindly advise. We have revised it.

Line 306:

Point 7: Figure 4 contain too small letters and digits.

Response 7: Thanks for your kindly suggestion, we have revised Figure 4 in the revised paper.

Line 244-246:

Point 8: The discussion in line 280 – 296 is very complicated especially because the authors mistook the parameter symbols and used some cited data very selectively.

Response 8: Thanks for your kindly comment. We have revised the parameter symbols and re-discussed the mechanical properties and degradability in strong acid.

Line 268-278: “Though there has a huge difference in the intrinsic viscosity between PHF-1 (0.230 dL/g) and PHF-6 (0.736 dL/g) due to the difference condensation temperature (Figure 2), they have a similar average Young’s modulus (E), maximum tensile strength (σm) and elongation at break (εb) from Figure 5(b), which was mainly due to the difference of their crystallinity. From the XRD results show in Figure 4 we can find that PHF-1 is a semi-crystalline polyester and PHF-6 is a crystalline polyester, which resulted in a similar mechanical property even they had a huge difference in the intrinsic viscosity. However, the E and σm obviously increased to 479 MPa and 36.5 MPa from about 450 MPa and 34 MPa with the addition of triphenyl phosphite, though there was a slight decrease in εb. Therefore, the addition of triphenyl phosphite improved the degree of polymerization of PHFs, thus enhancing its mechanical properties, which can be used for the preparation of high elongation fibre.”

Line 317-320: “In addition, we can find that the acid degradation rate of PHF-6 was lower than that of PHF-7 although it has a lower crystallinity than that of PHF-7, this was mainly due to the higher melting temperature of PHF-6 (Tm=145.3 °C, Figure 6) than that of PHF-7 (Tm=144.2 °C, Figure 6).”

Point 9: The synthesis of the polymers (PHFs preparation) is not described properly - no quantities of reagents were given polymers preparation selectively not seliymer kule properly quantities

Response 9: Thanks. We have added more information, and the revisions have been made accordingly, as follows:

Line 96-99: PHFs were synthesized via direct esterification method, which was performed as follows: 1) a mixture of HDO (0.01-0.03mol), FDCA (0.01mol), tetrabutyl titanate (0.03mmol) and triphenyl phosphite (0.05-0.5mmol) was loaded into a 50 mL three-neck round-bottomed flask, which was sealed and purged with N2 three times.

Point 10: Line 316-317 - authors observed signals derived from chains of more than six carbon  - is it really such chain in obtained polymers?

Response 10: Thanks for your kindly question, we have consulted the relevant textbooks and confirmed it.

Point 11: Line 331-332 - the degradation has not been discussed yet up to this part.

Response 11: Thanks for your kindly suggestion, we modified the description.

Line 310-329: The degradability of the three PHF membranes was determined by monitoring the weight loss with time under strong acid conditions, and the results are shown in Figure 7. The weight loss of the three PHF membranes is between 6.4-9.4 wt% when treated with strong acid for 1 week. The weight loss gradually increased with the increase of the treatment time, and the maximum weight loss reached about 20 wt% after 4 weeks treatment. As has been discussed in the part of 3.5, PHF-1 was a semi-crystalline polyester, and PHF-6 and PHF-7 were crystalline polyesters. And their crystallinity was 53.3%, 94.3% and 96.7%, respectively. As a result, it can be found from Figure 7(a) that the acid degradation rate of PHF-1 was higher than that of PHF-6 and PHF-7. In addition, we can find that the acid degradation rate of PHF-6 was lower than that of PHF-7 although it has a lower crystallinity than that of PHF-7, this was mainly due to the higher melting temperature of PHF-6 (Tm=145.3 °C, Figure 6) than that of PHF-7 (Tm=144.2 °C, Figure 6). As a result, the weight loss of PHF-1 (23.0 wt%) was higher than that of PHF-7 (20.9 wt%), and PHF-7 was higher that of PHF-6 (19.8 wt%). Compared with the FDCA-based polyesters reported in the published literature47, the weight loss of PHFs was obviously higher than that of the reported results, which only were 1 wt%, 2 wt% and 10 wt% in the fourth week, and 2 wt%, 5 wt% and 28 wt% in twenty-second week respectively under neutral, PH=4.0 and PH=12.0 conditions. It shows in Figure 7(a) that the weight loss is not linearly related to acid treatment time, which may be due to the difference in crystalline and amorphous regions.48 The environmental condition provided in this paper was harsher than that of the reported, but the degradation rate of the polyester was obviously higher, which exhibited better acid degradation performance.

Point 12: Line 329-331 - How the authors explain the mechanical properties of PHF1 and PHF6 and PHF7 samples based on the observed crystallinity (PHF6 and PHF7 are crystalline but the Young`s modulus is lower than the semi crystalline PHF1 – another parameters are almost the same).

Response 12: Thanks for your kindly comment. We have re-discussed it in the revised paper.

Line 268-278: Though there has a huge difference in the intrinsic viscosity between PHF-1 (0.230 dL/g) and PHF-6 (0.736 dL/g) due to the difference condensation temperature (Figure 2), they have a similar average Young’s modulus (E), maximum tensile strength (σm) and elongation at break (εb) from Figure 5(b), which was mainly due to the difference of their crystallinity. From the XRD results show in Figure 4 we can find that PHF-1 is a semi-crystalline polyester and PHF-6 is a crystalline polyester, which resulted in a similar mechanical property even they had a huge difference in the intrinsic viscosity. However, the E and σm obviously increased to 479 MPa and 36.5 MPa from about 450 MPa and 34 MPa with the addition of triphenyl phosphite, though there was a slight decrease in εb. Therefore, the addition of triphenyl phosphite improved the degree of polymerization of PHFs, thus enhancing its mechanical properties, which can be used for the preparation of high elongation fibre.

Point 13: The discussion on thermal properties of obtained polymers in line 347 – 359 is completely incomprehensible and inconsistent with the data in Figure 5.

-        In fig 5c the PHF1 sample has the highest decomposition temperature but in the fig 5b the temperature is lower.

-        How was the thermal history of samples investigated by the DSC method on the fig 5a (is it the first run or second or after quenching). In the experimental part the preparation is described very precisely but there is no information which run is presented in the fig 5a.

-        The sample PHF7 on the fig. 5b lost about 5% of mass already at a temperature of 250 °C, unlike the samples PHF1 and PHF6. This is not mentioned and not discussed in the text.

Response 13: Thanks for your kindly comment. We have made a mistake in Figure 6(c) and revised it in the revised paper. And the mass loss of PHF-7 at 100 °C has given the explain.

Line 293-295: Furthermore, the thermal stability of the PHFs was studied and the results are shown in Figure 6(b-c), and an about 5% mass loss of PHF-7 at 100 °C should be the carried moisture during the operation.

Point 14: How consistent are the results of mechanical and thermal tests - were the samples prepared in the same way for mechanical and thermal tests? Data from DSC and XRD tests (if they are true) should be consistent and confirm mechanical properties.

The discussion of the degradability of the obtained polymers in the strong acidic conditions is very intricate. The authors chose an extremely acid environment without giving reasons for their choice. Comparing the degradation of the polymers obtained with the described degradation experiments of other polymers under completely different conditions does not lead to any conclusions. The reference 46 is related to enzymatic degradation and therefore it is not suitable for this discussion.

Response 14: Thanks for your kindly comment. We have adjusted the structure of this paper and reexplained the mechanical property, thermal property and degradability in the revised paper.

Point 14: Line 366 – “weightlessness” means the lack of gravity and not the loss of weight

Response 14: Thank you for your suggestion. This has been changed to weight loss.

Point 15: The ref. 38 is cited not properly since “van” is only part of the surname (van Lith).

Response 10: Thanks for your kindly suggestion. This reference has been changed to another one according to Point 5.
